# Blossom Tree Graphical Models

**Zhe Liu**
Department of Statistics
University of Chicago

**John Lafferty**
Department of Statistics
Department of Computer Science
University of Chicago

## Abstract

We combine the ideas behind trees and Gaussian graphical models to form a new nonparametric family of graphical models. Our approach is to attach nonparanormal "blossoms", with arbitrary graphs, to a collection of nonparametric trees. The tree edges are chosen to connect variables that most violate joint Gaussianity. The non-tree edges are partitioned into disjoint groups, and assigned to tree nodes using a nonparametric partial correlation statistic. A nonparanormal blossom is then "grown" for each group using established methods based on the graphical lasso. The result is a factorization with respect to the union of the tree branches and blossoms, defining a high-dimensional joint density that can be efficiently estimated and evaluated on test points. Theoretical properties and experiments with simulated and real data demonstrate the effectiveness of blossom trees.

## 1 Introduction

Let $p^*(x)$ be a probability density on $\mathbb{R}^d$ corresponding to a random vector $X = (X_1, \ldots, X_d)$. The undirected graph $G = (V, E)$ associated with $p^*$ has $d = |V|$ vertices corresponding to $X_1, \ldots, X_d$, and missing edges $(i, j) \notin E$ whenever $X_i$ and $X_j$ are conditionally independent given the other variables. The undirected graph is a useful way of exploring and modeling the distribution.

In this paper we are concerned with building graphical models for continuous variables, under weaker assumptions than those imposed by existing methods. If $p^*(x) > 0$ is strictly positive, the Hammersley-Clifford theorem implies that the density has the form

$$p^*(x) \propto \prod_{C \in \mathcal{C}} \psi_C(x_C) = \exp\left(\sum_{C \in \mathcal{C}} f_C(x_C)\right). \tag{1.1}$$

In this expression, $\mathcal{C}$ denotes the set of cliques in the graph, and $\psi_C(x_C) = \exp(f_C(x_C)) > 0$ denotes arbitrary *potential functions*. This represents a very large and rich set of nonparametric graphical models. The fundamental difficulty is that it is in general intractable to compute the normalizing constant. A compromise must be made to achieve computationally tractable inference, typically involving strong assumptions on the functions $f_C$, on the graph $G = \{\mathcal{C}\}$, or both.

The default model for graphical modeling of continuous data is the multivariate Gaussian. When the Gaussian has covariance matrix $\Sigma$, the graph is encoded in the sparsity pattern of the precision matrix $\Omega = \Sigma^{-1}$. Specifically, edge $(i, j)$ is missing if and only if $\Omega_{ij} = 0$. Recent work has focused on sparse estimates of the precision matrix [8, 10]. In particular, an efficient algorithm for computing the estimator using a graphical version of the lasso is developed in [3]. The nonparanormal [5], a form of Gaussian copula, weakens the Gaussian assumption by imposing Gaussianity on the transformed random vector $f(X) = (f_1(X_1), \ldots, f_d(X_d))$, where each $f_j$ is a monotonic function. This allows arbitrary single variable marginal probability distributions in the model [5].

Both the Gaussian graphical model and the nonparanormal maintain tractable inference without placing limitations on the independence graph. But they are limited in their ability to flexibly model the bivariate and higher order marginals. At another extreme, forest-structured graphical models permit arbitrary bivariate marginals, but maintain tractability by restricting to acyclic graphs. An nonparametric approach based on forests and trees is developed in [7] as a nonparametric method for estimating the density in high-dimensional settings. However, the ability to model complex independence graphs is compromised.

In this paper we bring together the Gaussian, nonparanormal, and forest graphical models, using what we call blossom tree graphical models. Informally, a blossom tree consists of a forest of trees, and a collection of subgraphs–the blossoms—possibly containing many cycles. The vertex sets of the blossoms are disjoint, and each blossom contains at most one node of a tree. We estimate nonparanormal graphical models over the blossoms, and nonparametric bivariate densities over the branches (edges) of the trees. Using the properties of the nonparanormal, these components can be combined, or factored, to give a valid joint density for $X = (X_1, \ldots, X_d)$. The details of our construction are given in Section 2. We develop an estimation procedure for blossom tree graphical models, including an algorithm for selecting tree branches, partition the remaining vertices into potential blossoms, and then estimating the graphical structures of the blossoms. Since an objective is to relax the Gaussian assumption, our criterion for selecting tree branches is deviation from Gaussianity. Toward this end, we use the negentropy, showing that it has strong statistical properties in high dimensions. In order to partition the nodes into blossoms, we employ a nonparametric partial correlation statistic. We use a data-splitting scheme to select the optimal blossom tree structure based on held-out risk.

In the following section, we present the details of our method, including definitions of blossom tree graphs, the associated family of graphical models, and our estimation methods. In Sections 3 and 4, we present experiments with simulated and real data. Finally, we conclude in Section 5. Statistical properties, detailed proofs, and further experimental results are collected in a supplement.

## 2   Blossom Tree Graphs and Estimation Methods

To unify the Gaussian, nonparanormal and forest graphical models we make the following definition.

**Definition 2.1.** *A **blossom tree** on a node set $V = \{1, 2, \ldots, d\}$ is a graph $G = (V, E)$, together with a decomposition of the edge set $E$ as $E = F \cup \{\cup_{B \in \mathcal{B}} B\}$ satisfying the following properties:*

1. *$F$ is acyclic;*
2. *$V(B) \cap V(B') = \varnothing$, for $B, B' \in \mathcal{B}$ with $B \neq B'$, where $V(B)$ denotes the vertex set of $B$.*
3. *$|V(B) \cap V(F)| \leq 1$ for each $B \in \mathcal{B}$;*
4. *$V(F) \cup \bigcup_{\mathcal{B}} V(B) = V$.*

*The subgraphs $B \in \mathcal{B}$ are called **blossoms**. The unique node $\rho(B) \in V(B) \cap V(F)$, which may be empty, is called the **pedicel** of the blossom. The set of pedicels is denoted $\mathcal{P}(F) \subset V(F)$.*

Property 1 says that the set of edges $F$ forms a union of trees—a forest. Property 2 says that distinct blossoms share no vertices or edges in common. Property 3 says that each blossom is connected to at most one tree node. Property 4 says that every node in the graph is either in a tree or a blossom. Note that the blossoms are not required to be connected, but must have at most one vertex in common with the forest—this is the pedicel node.

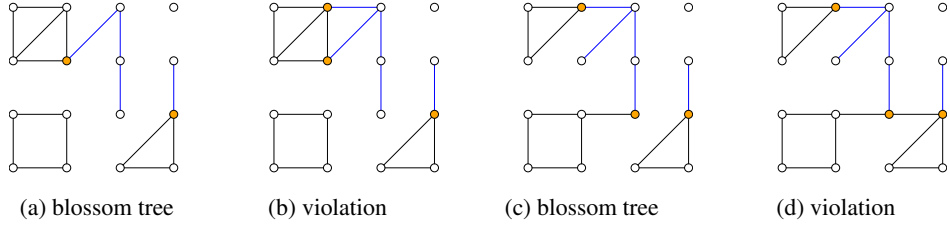

| (a) blossom tree | (b) violation | (c) blossom tree | (d) violation |

Figure 1: Four graphs, two blossom trees. The tree edges are colored blue, the blossom edges are colored black, and pedicels are orange. Graphs (a) and (c) correspond to blossom trees. Graphs (b) and (d) violate the restriction that each blossom has only a single pedicel, or attachment to a tree.

Suppose that $p(x) = p(x_1, \ldots, x_d)$ is the density of a distribution that has an independence graph given by a blossom tree $F \cup \{\cup_{\mathcal{B}} B\}$. Then from the blossom tree properties we have that

$$p(x) = p(X_{V(F)}) \prod_{B \in \mathcal{B}} p(X_{V(B)} \mid X_{V(F)}) \tag{2.1}$$

$$= p(X_{V(F)}) \prod_{B \in \mathcal{B}} p(X_{V(B)} \mid X_{\rho(B)}) \tag{2.2}$$

$$= p(X_{V(F)}) \prod_{B \in \mathcal{B}} \frac{p(X_{V(B)})}{p(X_{\rho(B)})} \tag{2.3}$$

$$= \prod_{(s,t) \in F} \frac{p(X_s, X_t)}{p(X_s)p(X_t)} \prod_{s \in V(F)} p(X_s) \prod_{B \in \mathcal{B}} \frac{p(X_{V(B)})}{p(X_{\rho(B)})} \tag{2.4}$$

$$= \prod_{(s,t) \in F} \frac{p(X_s, X_t)}{p(X_s)p(X_t)} \prod_{s \in V(F) \setminus \mathcal{P}(F)} p(X_s) \prod_{B \in \mathcal{B}} p(X_{V(B)}). \tag{2.5}$$

The first equality follows from disjointness of the blossoms. The second equality follows from the existence of a single pedicel node attaching the blossom to a tree. The fourth equality follows from the standard factorization of forests, and the last equality follows from the fact that each non-empty pedicel for a blossom is unique. We call the set of distributions that factor in this way the family of *blossom tree graphical models*.

A key property of the nonparanormal [5] is that the single node marginal probabilities $p(X_s)$ are arbitrary. This property allows us to form graphical models where each blossom distribution satisfies $X_{V(B)} \sim NPN(\mu_B, \Sigma_B, f_B)$, while enforcing that the single node marginal of the pedicel $\rho(B)$ agrees with the marginals of this node defined by the forest. This allows us to define and estimate distributions that are consistent with the factorization (2.5).

Let $X^{(1)}, \ldots, X^{(n)}$ be $n$ i.i.d. $\mathbb{R}^d$-valued data vectors sampled from $p^*(x)$ where $X^{(l)} = (X_1^{(l)}, \ldots, X_d^{(l)})$. Our goal is to derive a method for high-dimensional undirected graph estimation and density estimation, using a family of semiparametric estimators based on the blossom tree structure. Let $F_B$ denote the blossom tree structure $F \cup \{\cup_{\mathcal{B}} B\}$. Our estimation procedure is the following.

First, randomly partition the data $X^{(1)}, \ldots, X^{(n)}$ into two sets $\mathcal{D}_1$ and $\mathcal{D}_2$ of sample size $n_1$ and $n_2$. Then apply the following steps.

1. Using $\mathcal{D}_1$, estimate the bivariate densities $p^*(x_i, x_j)$ using kernel density estimation. Also, estimate the covariance $\Sigma^{ij}$ for each pair of variables. Apply Kruskal's algorithm on the estimated pairwise negentropy matrix to construct a family of forests $\{\widehat{F}^{(k)}\}$ with $k = 0, \ldots, d-1$ edges;

2. Using $\mathcal{D}_1$, for each forest $\widehat{F}^{(k)}$ obtained in Step 1, build the blossom tree-structured graph $\widehat{F}_{\widehat{B}}^{(k)}$. The forest structure $\widehat{F}^{(k)}$ is modeled by nonparametric kernel density estimators, while each blossom $\widehat{B}_i^{(k)}$ is modeled by the graphical lasso or nonparanormal. A family of

graphs is obtained by computing regularization paths for the blossoms, using the graphical lasso.

3. Using $\mathcal{D}_2$, choose $\widehat{F}_{\widehat{B}}^{(\widehat{k})}$ from this family of blossom tree models that maximizes the held-out log-likelihood.

The details of each step are presented below.

## 2.1 Step 1: Construct A Family of Forests

In information theory and statistics, negentropy is used as a measure of distance to normality. The negentropy is zero for Gaussian densities and is always nonnegative. The negentropy between variables $X_i$ and $X_j$ is defined as

$$J(X_i; X_j) = H(\phi(x_i, x_j)) - H(p^*(x_i, x_j)), \tag{2.6}$$

where $H(\cdot)$ denotes the differential entropy of a density, and $\phi(x_i, x_j)$ is an Gaussian density with the same mean and covariance matrix as $p^*(x_i, x_j)$.

Kruskal's algorithm [4] is a greedy algorithm to find a maximum weight spanning tree of a weighted graph. At each step it includes an edge connecting the pair of nodes with the maximum weight among all unvisited pairs, if doing so does not form a cycle. The algorithm also results in the best $k$-edge weighted forest after $k < d$ edges have been included.

In our setting, we define the weight $w(i, j)$ of nodes $i$ and $j$ as the negentropy between $X_i$ and $X_j$, and use Kruskal's algorithm to build the maximum weight spanning forest $\widehat{F}^{(k)}$ with $k$ edges where $k < d$. In such a way, the pairs of nodes that are less likely to be a bivariate Gaussian are included in the forest and then are modeled nonparametrically.

Since the true density $p^*$ is unknown, we replace the population negentropy $J(X_i; X_j)$ by the estimate

$$\widehat{J}_{n_1}(X_i; X_j) = H(\widehat{\phi}_{n_1}(x_i, x_j)) - \widehat{H}(\widehat{p}_{n_1}(x_i, x_j)), \tag{2.7}$$

where $\widehat{\phi}_{n_1}(x_i, x_j)$ is an estimate of the Gaussian density $\phi(x_i, x_j)$ for $X_i$ and $X_j$ using $\mathcal{D}_1$, $\widehat{p}_{n_1}(x_i, x_j)$ is a bivariate kernel density estimate for $X_i$ and $X_j$, and $\widehat{H}(\cdot)$ denotes the empirical differential entropy. In particular, let $\Sigma^{ij}$ be the covariance matrix of $X_i$ and $X_j$. Denote $\widehat{\Sigma}_{n_1}^{ij}$ as the empirical covariance matrix of $X_i$ and $X_j$ based on $\mathcal{D}_1$, then the plug-in estimate

$$H(\widehat{\phi}_{n_1}(x_i, x_j)) = 1 + \log(2\pi) + \frac{1}{2}\text{logdet}(\widehat{\Sigma}_{n_1}^{ij}). \tag{2.8}$$

Let $K(\cdot)$ be a univariate kernel function. Then given an evaluation point $(x_i, x_j)$, the bivariate kernel density estimate for $(X_i, X_j)$ based on observations $\{X_i^{(l)}, X_j^{(l)}\}_{l \in \mathcal{D}_1}$ is given by

$$\widehat{p}_{n_1}(x_i, x_j) = \frac{1}{n_1} \sum_{l \in \mathcal{D}_1} \frac{1}{h_{2i}h_{2j}} K\left(\frac{X_i^{(l)} - x_i}{h_{2i}}\right) K\left(\frac{X_j^{(l)} - x_j}{h_{2j}}\right), \tag{2.9}$$

where $h_{2i}$ and $h_{2j}$ are bandwidth parameters for $(X_i, X_j)$. To compute the empirical differential entropy $\widehat{H}(\widehat{p}_{n_1}(x_i, x_j))$, we numerically evaluate a two-dimensional integral.

Once the estimated negentropy matrix $\left[\widehat{J}_{n_1}(X_i; X_j)\right]_{d \times d}$ is obtained, we apply Kruskal's algorithm to construct a family of forests $\{\widehat{F}^{(k)}\}_{k=0...d-1}$.

## 2.2 Step 2: Build and Model the Blossom Tree Graphs

Suppose that we have a forest-structured graph $F$ with $|V(F)| < d$ vertices. Then for each remaining non-forest node, we need to determine which blossom it belongs to. We exploit the following basic fact.

**Proposition 2.1.** *Suppose that $X \sim p^*$ is a density for a blossom tree graphical model with forest $F$. Let $i \notin V(F)$ and $s \in V(F)$. Then node $i$ is not in a blossom attached to tree node $s$ if and only if*

$$X_i \perp\!\!\!\perp X_s \mid X_t \quad \text{for some node } t \in V(F) \text{ such that } (s,t) \in E(F). \tag{2.10}$$

We use this property, together with a measure of partial correlation, in order to partition the non-forest nodes into blossoms. Partial correlation measures the degree of association between two random variables, with the effect of a set of controlling random variables removed. Traditionally, the partial correlation between variables $X_i$ and $X_s$ given a controlling variable $X_t$ is the correlation between the residuals $\epsilon_{i \setminus t}$ and $\epsilon_{s \setminus t}$ resulting from the linear regression of $X_i$ with $X_t$ and of $X_s$ with $X_t$, respectively. However, if the underlying joint Gaussian or nonparanormal assumption is not satisfied, linear regression cannot remove all of the effects of the controlling variable.

We thus use a nonparametric version of partial correlation. Following [1], suppose $X_i = g(X_t) + \epsilon_{i \setminus t}$ and $X_s = h(X_t) + \epsilon_{s \setminus t}$, for certain functions $g$ and $h$ such that $\mathbb{E}(\epsilon_{i \setminus t} \mid X_t) = 0$ and $\mathbb{E}(\epsilon_{s \setminus t} \mid X_t) = 0$. Define the *nonparametric partial correlation* as

$$\rho_{is \cdot t} = \mathbb{E}(\epsilon_{i \setminus t} \epsilon_{s \setminus t}) \Big/ \sqrt{\mathbb{E}(\epsilon_{i \setminus t}^2) \, \mathbb{E}(\epsilon_{s \setminus t}^2)}. \tag{2.11}$$

It is shown in [1] that if $X_i \perp\!\!\!\perp X_s \mid X_t$, then $\rho_{is \cdot t} = 0$. We thus conclude the following.

**Proposition 2.2.** *If $\rho_{is \cdot t} \neq 0$ for all $t$ such that $(s,t) \in E(F)$, node $i$ is in a blossom attached to node $s$.*

Let $\widehat{g}$ and $\widehat{h}$ be local polynomial estimators of $g$ and $h$, and $\widehat{\epsilon}_{i \setminus t}^{(l)} = X_i^{(l)} - \widehat{g}(X_t^{(l)})$, $\widehat{\epsilon}_{s \setminus t}^{(l)} = X_s^{(l)} - \widehat{h}(X_t^{(l)})$ for any $l \in \mathcal{D}_1$, then an estimate of $\rho_{is \cdot t}$ is given by

$$\widehat{\rho}_{is \cdot t} = \sum_{l \in \mathcal{D}_1} (\widehat{\epsilon}_{i \setminus t}^{(l)} \, \widehat{\epsilon}_{s \setminus t}^{(l)}) \Big/ \sqrt{\sum_{l \in \mathcal{D}_1} (\widehat{\epsilon}_{i \setminus t}^{(l)})^2 \sum_{l \in \mathcal{D}_1} (\widehat{\epsilon}_{s \setminus t}^{(l)})^2}. \tag{2.12}$$

Based on Proposition 2.2, for each forest $\widehat{F}^{(k)}$ obtained in Step 1, we then assign each non-forest node $i$ to the blossom with the pedicel given by

$$\widehat{s}_i = \operatorname*{argmax}_{s \in V(\widehat{F}^{(k)})} \min_{\{t \colon (s,t) \in E(\widehat{F}^{(k)})\}} |\widehat{\rho}_{is \cdot t}|. \tag{2.13}$$

After iterating over all non-forest nodes, we obtain a blossom tree-structured graph $\widehat{F}_{\widehat{B}}^{(k)}$. Then the forest structure is nonparametrically modeled by the bivariate and univariate kernel density estimations, while each blossom is modeled with the graphical lasso or nonparanormal. In particular, when $k = 0$ that there is no forest node, our method is reduced to modeling the entire graph by the graphical lasso or nonparanormal.

Alternative testing procedures based on nonparametric partial correlations could be adopted for partitioning nodes into blossoms. However, such methods may have large computational cost, and low power for small sample sizes.

Note that while each non-forest node is associated with a pedicel in this step, after graph estimation for the blossoms, the node may well become disconnected from the forest.

## 2.3   Step 3: Optimize the Blossom Tree Graphs

The full blossom tree graph $\widehat{F}_{\widehat{B}}^{(d-1)}$ obtained in Steps 1 and 2 might result in overfitting in the density estimate. Thus we need to choose an optimal graph with the number of forest edges $k \leq d - 1$. Besides, the tuning parameters involved in the fitting of each blossom by the graphical lasso or nonparanormal also induce a bias-variance tradeoff.

To optimize the blossom tree structures over $\{\widehat{F}_{\widehat{B}}^{(k)}\}_{k=0\ldots d-1}$, we choose the complexity parameter $\widehat{k}$ as the one that maximizes the log-likelihood on $\mathcal{D}_2$, using the factorization (2.5):

$$\widehat{k} = \operatorname*{argmax}_{k\in\{0,\ldots,d-1\}} \frac{1}{n_2} \sum_{l\in\mathcal{D}_2} \log \left( \prod_{(i,j)\in E(\widehat{F}^{(k)})} \frac{\widehat{p}_{n_1}(X_i^{(l)}, X_j^{(l)})}{\widehat{p}_{n_1}(X_i^{(l)})\widehat{p}_{n_1}(X_j^{(l)})} \cdot \right.$$

$$\left. \prod_{s\in V(\widehat{F}^{(k)})\setminus\mathcal{P}(\widehat{F}^{(k)})} \widehat{p}_{n_1}(X_s^{(l)}) \prod_{i=1}^{k} \widehat{\phi}_{n_1}^{\lambda_i^{(k)}} \left( X_{V(\widehat{B}_i^{(k)})}^{(l)} \right) \right), \tag{2.14}$$

where $\widehat{\phi}_{n_1}^{\lambda_i^{(k)}}$ is the density estimate for blossoms by the graphical lasso or nonparanormal, with the tuning parameter $\lambda_i^{(k)}$ selected to maximize the held-out log-likelihood. That is, the complexity of each blossom is also optimized on $\mathcal{D}_2$.

Thus the final blossom tree density estimator is given by

$$p_{\widehat{F}_{\widehat{B}}^{(\widehat{k})}}(x) = \prod_{(i,j)\in E(\widehat{F}^{(\widehat{k})})} \frac{\widehat{p}_{n_1}(x_i, x_j)}{\widehat{p}_{n_1}(x_i)\widehat{p}_{n_1}(x_j)} \prod_{s\in V(\widehat{F}^{(\widehat{k})})\setminus\mathcal{P}(\widehat{F}^{(\widehat{k})})} \widehat{p}_{n_1}(X_s^{(l)}) \prod_{i=1}^{\widehat{k}} \widehat{\phi}_{n_1}^{\lambda_i^{(\widehat{k})}}(x_{\widehat{B}_i^{(\widehat{k})}}). \tag{2.15}$$

In practice, Step 3 can be carried out simultaneously with Steps 1 and 2. Whenever a new edge is added to the current forest in Kruskal's algorithm, the blossoms are re-constructed and re-modeled. Then the held-out log-likelihood of the obtained density estimator can be immediately computed. In addition, since there are no overlapping nodes between different blossoms, the sparsity tuning parameters are selected separately for each blossom, which reduces the computational cost considerably.

# 3   Analysis of Simulated Data

Here we present numerical results based on simulations. We compare the blossom tree density estimator with the graphical lasso [3] and forest density estimator [7]. To evaluate the performance of these estimators, we compute and compare the log-likelihood of each method on held-out data.

We simulate high-dimensional data which are consistent with an undirected graph. We generate multivariate non-Gaussian data using a sequence of mixtures of two Gaussian distributions with contrary correlation and equal weights. Then a subset of variables are chosen to generate the blossoms that are distributed as multivariate Gaussians. In dimensional $d = 80$, we sample $n_1 = n_2 = 400$ data points from this synthetic distribution.

A typical run showing the held-out log-likelihood and estimated graphs is provided in Figures 2 and 3. The term "trunk" is used to represent the edge added to the forest in a blossom tree graph. We can see that the blossom tree density estimator is superior to other methods in terms of generalization performance. In particular, the graphical lasso is unable to uncover the edges that are generated nonparametrically. This is expected, since different blossoms have zero correlations among each other and are thus regarded as independent by the algorithm of graphical lasso. For the modeling of the variables that are contained in a blossom and are thus multivariate Gaussian distributed, there is an efficiency loss in the forest density estimator, compared to the graphical lasso. This illustrates the advantage of blossom tree density estimator. As is seen from the number of selected edges by each method shown in Figure 2, the blossom tree density estimator selects a graph with a similar sparsity pattern as the true graph.

# 4   Analysis of Cell Signalling Data

We analyze a flow cytometry dataset on $d = 11$ proteins from [9]. A subset of $n = 853$ cells were chosen. A nonparanormal transformation was estimated and the observations, for each variable,

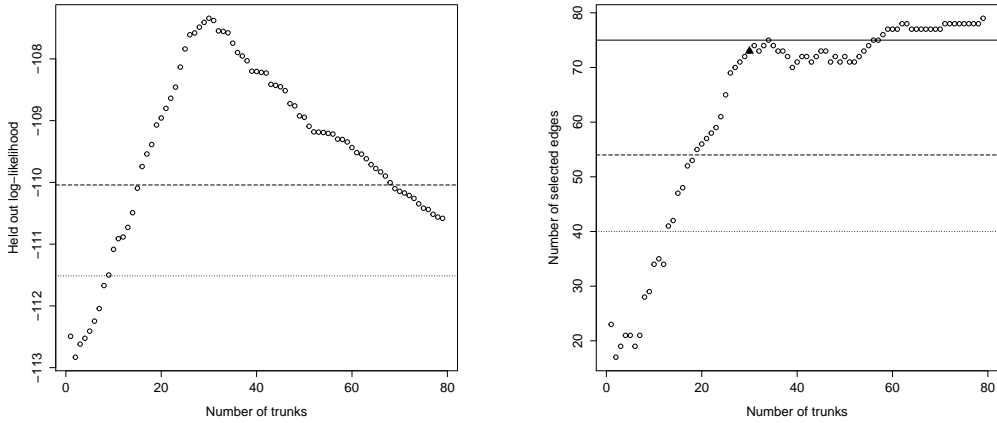

Figure 2: Results on simulations. Left: Held-out log-likelihood of the graphical lasso (horizontal dotted line), forest density estimator (horizontal dashed line), and blossom tree density estimator (circles); Right: Number of selected edges by these methods. The horizontal solid line indicates the number of edges in the true graph, and the solid triangle indicates the best blossom tree graph. The first circle for blossom tree refers to the 1-trunk case.

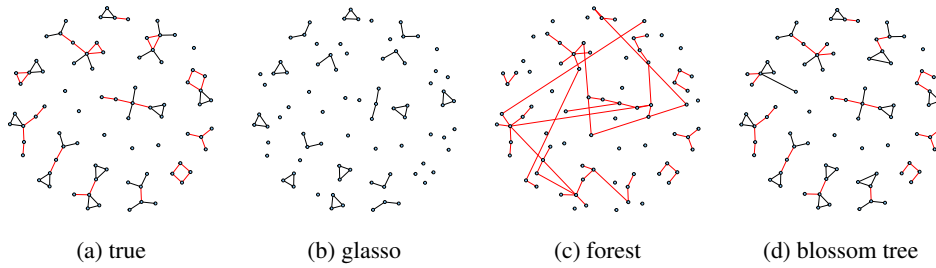

|                 |                 |                  |                       |
|:---------------:|:---------------:|:----------------:|:---------------------:|
| (a) true        | (b) glasso      | (c) forest       | (d) blossom tree      |

Figure 3: Results on simulations. Graph (a) corresponds to the true graph. Graphs (b), (c) and (d) correspond to the estimated graphs by the graphical lasso, forest density estimator, and blossom tree density estimator, respectively. The tree edges are colored red, and the blossom edges are colored black.

were replaced by their respective normal scores, subject to a Winsorized truncation [5]. We study the associations among the proteins using the graphical lasso, forest density estimator, and blossom tree forest density estimator. The maximum held-out log-likelihood for glasso, forest, and blossom tree are -14.3, -13.8, and -13.7, respectively. We can see that blossom tree is slighter better than forest in terms of the generalization performance, both of which outperform glasso. Results of estimated graphs are provided in Figures 4. When the maximum of held-out log-likelihood curve is achieved, glasso selects 28 edges, forest selects 7 edges, and blossom tree selects 10 edges. The two graphs uncovered by forest and blossom tree agree on most edges, although the latter contains cycles.

# 5   Conclusion

We have proposed a combination of tree-based graphical models and Gaussian graphical models to form a new nonparametric approach for high dimensional data. Blossom tree models relax the normality assumption and increase statistical efficiency by modeling the forest with nonparametric kernel density estimators and modeling each blossom with the graphical lasso or nonparanormal. Our experimental results indicate that this method can be a practical alternative to standard approaches to graph and density estimation.

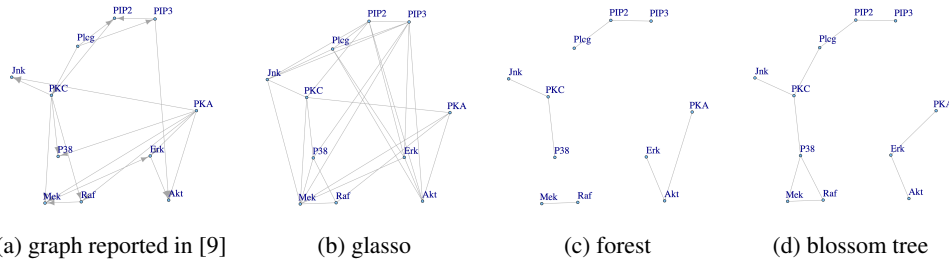

|  (a) graph reported in [9] | (b) glasso | (c) forest | (d) blossom tree |

Figure 4: Results on cell signalling data. Graph (a) refers to the fitted graph reported in [9]. Graphs (b), (c) and (d) correspond to the estimated graphs by the graphical lasso, forest density estimator, and blossom tree density estimator, respectively.

# Acknowledgements

Research supported in part by NSF grant IIS-1116730, AFOSR grant FA9550-09-1-0373, ONR grant N000141210762, and an Amazon AWS in Education Machine Learning Research grant.

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
