[Supplementary Material]

# A  Supplement

**Statistical Properties.**

Here we discuss statistical properties of the estimates of negentropy and nonparametric partial correlation. For any pair of variables $X_i$ and $X_j$, let $(X_i^{(1)}, X_j^{(1)}), \ldots, (X_i^{(n)}, X_j^{(n)})$ be $n$ i.i.d. $\mathbb{R}^2$-valued data vectors sampled from $p^*(x_i, x_j)$. Suppose $\mathcal{X} = [0, 1]^2$ is the domain of $(X_i, X_j)$.

**Assumption A.1.** *For any pair of variables $X_i$ and $X_j$, assume the density $p^*(x_i, x_j)$ belongs to a 2nd-order Hölder class and is bounded away from zero and infinity.*

**Assumption A.2.** *For any sequence converging to a boundary point, suppose the density $p^*(x_i, x_j)$ of any pair of variables $X_i$ and $X_j$ has vanishing first order partial derivatives.*

**Assumption A.3.** *Assume the kernel function $K(\cdot)$ is nonnegative and has a bounded support $[-1, 1]$ with $\int_{-1}^{1} K(x)dx = 1$ and $\int_{-1}^{1} xK(x)dx = 0$.*

Under these assumptions, the authors in [6] prove an exponential concentration inequality based on a clipped "mirror image" kernel density estimator $\tilde{p}_{n,h}(x_i, x_j)$, where $h$ is the bandwidth parameter. Combined with bounds on the log determinant of sample covariance matrices presented in [2], we obtain the following.

**Theorem A.1.** *Under Assumptions A1-A3, for any $\epsilon > 0$, if we choose the bandwidth according to $h \asymp n^{-1/4}$, then there exists a constant $N$ such that if $n > N$,*

$$P(|\tilde{J}_n(X_i; X_j) - J(X_i; X_j)| > \epsilon) \leq 3 \exp\left(-\frac{n\epsilon^2}{144\kappa^2}\right),$$

*for any $i, j \in \{1, \ldots, d\}$ with $i \neq j$, where*

$$\tilde{J}_n(X_i; X_j) = 1 + \log(2\pi) + \frac{1}{2}\text{logdet}(\widehat{\Sigma}_n^{ij}) - \frac{3}{2(n-1)} - H(\tilde{p}_{n,h}(x_i, x_j)),$$

*and $\kappa = \max\{|\log(\min_{\mathcal{X}} p^*(x_i, x_j))|, |\log(\max_{\mathcal{X}} p^*(x_i, x_j))|\} + 1$.*

The advantage of having this exponential inequality is that we can guarantee accurate estimation of the negentropy for many pairs of variables simultaneously. It is easy to see that the above theorem implies a $\sqrt{n}$-rate of convergence in mean squared error $\mathbb{E}(|\tilde{J}_n(X_i; X_j) - J(X_i; X_j)|) = O(n^{-1/2})$.

To justify the estimation of nonparametric partial correlation, it is shown in [1] that, for some $p_1, p_2 > 0$,

$$\sqrt{n}(\widehat{\rho}_{is\cdot t} - \rho_{is\cdot t}) = \sqrt{n}(r_{is\cdot t} - \rho_{is\cdot t}) + O_P(n^{-\min\{p_1, p_2\}}),$$

where $r_{is\cdot t}$ is the empirical correlation coefficient based on the errors $\epsilon_{i\setminus t}^{(l)} = X_i^{(l)} - g(X_t^{(l)})$, $\epsilon_{s\setminus t}^{(l)} = X_s^{(l)} - h(X_t^{(l)})$ for any $l \in \mathcal{D}_1$, and $n^{p_1}(\widehat{g}(x) - g(x)) = O_P(1)$, $n^{p_2}(\widehat{h}(x) - h(x)) = O_P(1)$.

**Proof of Proposition 2.1.**

If node $i$ is not in a blossom attached to node $s$, then either node $i$ is not attached to any forest node (its pedicel is $\varnothing$), or there exists some forest node $t$ such that node $i$ belongs to the blossom attached to $t$. In either case, we have $X_i \perp\!\!\!\perp X_s \mid X_t$. For the other direction, assume on the contrary that node $i$ is in a blossom attached to node $s$. Then nodes $i$ and $s$ are not independent conditional on node $t$. $\square$

**Proof of Proposition 2.2.**

Suppose node $i$ is not in a blossom attached to the forest node $s$. Then according to Proposition 2.1, there exists some node $t$ such that $(s, t) \in E(F)$, satisfying $X_i \perp\!\!\!\perp X_s \mid X_t$. This implies $\rho_{is\cdot t} = 0$, contradicting the assumption. $\square$

**Proof of Theorem A.1.**

The authors in [6] show that under Assumptions A1-A3, for any $\epsilon > 0$, if we choose the bandwidth according to $h \asymp n^{-1/4}$, then if $n$ is large enough,

$$P(|H(\tilde{p}_n(x_i, x_j)) - H(p^*(x_i, x_j))| > \epsilon) \leq 2 \exp\left(-\frac{n\epsilon^2}{36\kappa^2}\right).$$

A limiting law of the log determinant of the sample covariance matrix is presented in [2], where as $n \to \infty$

$$\frac{\text{logdet}(\widehat{\Sigma}_n^{ij}) - 3/(n-1) - \text{logdet}(\Sigma_n^{ij})}{2/\sqrt{n-1}} \xrightarrow{L} \mathcal{N}(0, 1).$$

Then if $n$ is large enough, we obtain

$$P(|\text{logdet}(\widehat{\Sigma}_n^{ij}) - 3/(n-1) - \text{logdet}(\Sigma_n^{ij})| > \epsilon) \leq \sqrt{\frac{8}{(n-1)\pi\epsilon^2}} \exp\left(-\frac{(n-1)\epsilon^2}{8}\right).$$

Following from the union bound, there exists a constant $N$ such that for all $n > N$,

$$P(|\tilde{J}_n(X_i; X_j) - J(X_i; X_j)| > \epsilon) \leq \sqrt{\frac{8}{(n-1)\pi\epsilon^2}} \exp\left(-\frac{(n-1)\epsilon^2}{8}\right) + 2 \exp\left(-\frac{n\epsilon^2}{144\kappa^2}\right),$$

$$\leq 3 \exp\left(-\frac{n\epsilon^2}{144\kappa^2}\right).$$

$\square$

**Supplementary Simulations 1.**

We also evaluate the performance of these methods when the number of nodes within each blossom $|V(B_i)|$ varies. We generate non-Gaussian data for 5 variables with a Toeplitz dependence structure using a Student t-copula, and then simulate multivariate Gaussian data for the blossoms connected to each of these 5 nodes. The number of nodes within each blossom is set to be equal across different blossoms and varies from 0 to 6.

Table 1 compares the held-out log-likelihood for each method. We can see that when $|V(B_i)|$ is small, the forest density estimator has the best generalization performance. As $|V(B_i)|$ increases, the blossom tree density estimator dominates the other two methods. When $|V(B_i)|$ becomes large, the graphical lasso estimator performs better than the forest density estimator, although the blossom tree density estimator is still the best.

Table 1: Results on simulations. Held-out log-likelihood of graphical lasso, forest density estimator, and blossom tree density estimator as a function of the number of nodes within each blossom $|V(B_i)|$, which is set to be equal across different blossoms and varies from 0 to 6

| $|V(B_i)| =$ | 0 | 1 | 2 | 3 | 4 | 5 | 6 |
|---|---|---|---|---|---|---|---|
| glasso | -6.61 | -13.50 | -20.66 | -27.84 | -34.69 | -41.25 | -48.19 |
| forest | -6.11 | -13.04 | -20.19 | -27.35 | -34.55 | -41.23 | -48.42 |
| blossom tree | -6.32 | -13.08 | -20.05 | -27.21 | -34.20 | -40.79 | -47.84 |

**Supplementary Simulations 2.**

Suppose variables $X_s, X_t$ and $X_i$ satisfy the following distributions

$$(X_s, X_t) \sim \begin{cases} \mathcal{N}\left( \begin{pmatrix} 0 \\ 0 \end{pmatrix}, \begin{pmatrix} 1 & 0.8 \\ 0.8 & 1 \end{pmatrix} \right) & \text{with probability } 0.5 \\ \mathcal{N}\left( \begin{pmatrix} 0 \\ 0 \end{pmatrix}, \begin{pmatrix} 1 & -0.8 \\ -0.8 & 1 \end{pmatrix} \right) & \text{with probability } 0.5 \end{cases},$$

$$(X_i, X_s) \sim \mathcal{N}\left( \begin{pmatrix} 0 \\ 0 \end{pmatrix}, \begin{pmatrix} 1 & r \\ r & 1 \end{pmatrix} \right).$$

Assume a forest structure $F$ that only contains one edge $(s, t)$ between node $s$ and node $t$. We would like to evaluate the success probability that another node $i$ is assigned to the blossom attached to node $s$ rather than node $t$ using the procedure described in Section 2.

Figure 5 reports the proportions of times among 200 replications that node $i$ was classified to the blossom attached to forest node $s$ or $t$, as a function of the correlation coefficient $r$ or sample size. It is as expected that the probability that node $i$ is successfully assigned to the blossom connected to node $s$ becomes larger as the correlation or sample size increases.

Figure 5: Proportions of times among 200 replications that node $i$ was classified to the blossom attached to forest node $s$ or $t$, as a function of the correlation coefficient $r$ with sample size $n = 200$ (left), or as a function of sample size with correlation $r = 0.1$ (right)

**Supplementary Real Data Analysis.**

For the analysis of cell signalling data, Figure 6 shows the held-out log-likelihood of the graphical lasso, forest density estimator, and blossom tree density estimator.

Figure 6: Results on cell signalling data. Held-out log-likelihood of the graphical lasso (horizontal dotted line), forest density estimator (horizontal dashed line), and blossom tree density estimator (circles)