[Reviews · NeurIPS 2014]

Submitted by Assigned_Reviewer_5

Authors propose a method of estimating a graphical model for continuous data that blends the following three, established ideas: 1) assume the data follows a multivariate Gaussian and estimate using the graphical lasso; 2) do not assume the data follows a multivariate Gaussian and instead use a Gaussian copula, the nonparanormal, to allow arbitrary single variable marginals; or 3) assume a specific tree-structured factorization and model arbitrary bivariate marginals along the tree structure.

The proposed method introduces the blossom tree, which is a specific factorization of the model into a collection of densely connected blossom components that are connected by a specific set of tree edges. In particular, each blossom is connected (via a pedicel node) to at most one tree edge. The blossom components are modeled as sparse multivariate Gaussians (or using the non-paranormal copula) and the tree edges are modeled as arbitrary bivariate distributions with single variable marginals that are consistent with the marginal of any blossom pedicel to which they are attached.

The main idea of this paper, the factorization of a high-dimensional graphical model via the blossom tree structure appears to be unique and the ability to flexibly model components of a high-dimensional graphical model as non-Gaussian seems appealing. However, I’m not convinced that the blossom-tree factorization adds any real value. In other words, why should I do all of this work to learn a blossom-tree model, over a non-paranomral model, a sparse multivariate gaussian or a forest-tree model? The authors only considered simulated data, constructed using a process favorable for their proposed factorization.

I found the result in Theorem 3.1 a bit confusing. It tells us about the quality of estimating the negentropy. Why is this useful for the joint estimation problem? What does it tell us about the estimates produced using the blossom-tree factorization?

In addition, the proposed method of learning the blossom-tree structure seems a bit weak. Learning a sequence of blossom tree models, one non-blossom edge at a time, seems unreasonable for any moderately sized problem. In addition, please be specific about the complexity of the proposed method, relative to learning using graphical lasso, non paranormal and forest-trees.

Finally, I think the experiments section could use a real boost. Running on at least one real-world data set would help convince the reader that there is real value in this factorization.

Detailed comments:
- I think that calling this a blossom tree is a poor choice. The word blossom was introduced by Edmonds for the matching problem and refers to an odd-sized set of vertices.
- In the experiments, why are the ‘0’ trunks not equal to the graphical lasso? If you don’t add any trunks isn’t there just one blossom, modeled as a multivariate gaussian?
- Runtimes for the different methods should also be reported.
- Line 60: The statement “maintain tractable inference without placing limitations on the independence graph” is a bit misleading. They do encourage the graph to be sparse; they are just not structural like, say a graph laplacian.
Summary: Interesting article that is well written and appears to be introduce a novel factorization for graphical models of continuous data. However, the lack of experimentation on real-world data failed to convince me of the utility of the blossom tree factorization, while the proposed greed blossom tree construction method seems to intractable to be useful.

Submitted by Assigned_Reviewer_14

This paper proposes blending several methods for statistically modeling
multivariable continuous data into a method called blossom tree. The different methods include forest models and the graphical lasso. The resulting type of model seems novel and quite flexible and the proposed algorithm may be able to efficiently estimate it from high-dimensional non-Gaussian data. The paper is mostly clear but not easy to follow, in part because the way the model is constructed seems a bit ad-hoc, in part because many results are left for the supplementary material. It will be helpful to make the paper more principled and self-contained. Experimental results are shown only for simulated data that are constructed with the proposed model in mind. It would be useful to see a stronger experimental section including a substantial result(s) on a real dataset with e.g. glasso benchmark.
Summary: Mostly clearly written paper presenting a new and potentially useful method to flexibly model non-Gaussian multidimensional continuous data. The experimental demonstration is lacking.

Submitted by Assigned_Reviewer_21

The paper continues a line of work in graphical models to go beyond
Gaussian modeling of real-valued attributes. The apparent novelty is
the combination of a previous algorithm for non-parametric forest
density estimation, which rests on a non-parametric generalization of
the classical Chow-Liu algorithm for learning tree graphical models, with the standard
graphical LASSO (glasso), resulting on the authors call a "blossom tree
graphical model." The paper states a type of statistical-consistency
(theoretical) result, and a few experiments.

* On Section 3: Can the authors' comment on how much the stated result
says about *machine learning* properties (i.e., PAC or
large-deviation, as opposed to data-size-asymptotic bounds)? Can the
authors say anything at all about the properties of the constant N?
What would be an expression of a lower or upper bound, even if exact?
Just like the typical asymptotic convergence as the number of samples
goes to infinity (e.g., CLT), it is hard to know when the
large-deviation bounds of the kind provided in the paper start being
valid. Without such information regarding N and the exact number
of samples n, while it may be an interesting and useful statistical
result, it does not appear very useful to machine learning, IMHO.

Hence, I would have Section 3 moved fully to the supplementary
material and replace it with more experimental evaluations. For example,
I would have moved the sections Supplementary Simulations 1 and 2 into
the main body of the paper. But even then, a slightly more thorough
evaluation would be useful such as experimenting by changing the class
or form of the underlying ground-truth density, and the class of
underlying graphs, whenever applicable, the number of samples, and the
number of attributes/variables.

* Given the nature of the paper, a reference to the Chow-Liu
algorithm, from 1968, seems warranted, even if only in passing.

C. K. Chow, and C. N. Liu, "Approximating Discrete Probability
Distributions with Dependence Trees," IEEE Transactions of Information
Theory, vol. IT-14, no. 3, May 1968

Despite the different statistical setups/models (i.e., discrete
vs. continuous), and the fact that the work in the submitted paper
goes beyond simple trees/forests/blossom representations, it is
undeniable that the core of the underlying approach has its roots on
Chow-Liu's work.

* It'd have been nice to see experiments on real-world data such as
those on microarray data that Liu et al (JMLR, 2011) performed. Is
the problem public access to the data? What about other datasets
(e.g., fMRI)?

* I am curious as the core distinctions among the following work and
forest density estimation, which serves as the foundation for the submission?

- Marwan Mattar, and Erik Learned-Miller. Improved generative models for
continuous image features through tree-structured non-parametric
distributions. UMass Amherst Technical Report 06-57, 10 pages, 2006.

- Mihai Datcu, Farid Melgani, Andrea Piardi, and Sebastiano
B. Serpico, "Multisource Data Classification With Dependence Trees,"
IEEE Transactions on Geoscience and Remote Sensing, vol. 40, no. 3,
pp. 609-617, March 2002.

I am interested in directly ML-relevant, as opposed to statistical,
distinctions.

* It has been quite a long time since Chow and Liu's work in 1968. Hence,
I was originally skeptical that no one had actually tried anything like what the
authors' propose. If it exists, I have not been able to find it. So,
while somewhat surprising, I must give the authors their due for the
novelty and originality of their proposed approach... Now, I may
change my mind later, should I find a reference :) ... All kidding aside,
even if I find an old reference, it does not take much away from the
authors' originality/novelty, because no one in recent years have
proposed the technique, despite the intense attention to the problem
of modeling continuous/real-valued attributes going back a decade, at least!

Summary: A nice, clean, interesting, and surprisingly novel approach to
modeling multiple real-valued attributes/random variables. It can benefit
from further experimental/empirical evaluations.

Author Feedback
Author rebuttal: We thank the reviewers for their helpful comments.

As suggested by the reviewers, we conducted experiments on real-world data to evaluate the performance of our proposed method.

We analyzed the flow cytometry dataset on d = 11 proteins from Sachs et al. (2003). A subset of n = 853 cells were chosen. A nonparanormal transformation was estimated and the observations, for each variable, were replaced by their respective normal scores, subject to a Winsorized truncation. We studied the associations among the proteins using glasso, forest, and blossom tree. The maximum
heldout log-likelihood for glasso, forest, and blossom tree are -14.3, -13.8, and -13.7, respectively. We can see that blossom tree is slighter better than forest in terms of the generalization performance, both of which outperform the glasso. In particular, when a maximum of heldout log-likelihood curve is achieved, glasso selects 28 edges, the forest density selects 7 edges, and the blossom tree selects 10 edges. The two graphs uncovered by the forest and blossom tree agree on most edges, although the latter contains cycles.

We will include these results in a revision of the paper.

Some additional replies to reviewers' comments are below:

Reviewer 21:

> On Section 3: Can the authors' comment on how much the stated result says about *machine learning* properties (i.e., PAC or large-deviation, as opposed to data-size-asymptotic bounds)? Can the authors say anything at all about the properties of the constant N?

In fact, the result of Theorem 3.1 is a non-asymptotic, finite sample exponential concentration inequality (that is, a large-deviation inequality). The constant N depends on the lower and upper bounds on the density, and on the Holder constant assumed of the density.

> Given the nature of the paper, a reference to the Chow-Liu algorithm, from 1968, seems warranted, even if only in passing.

We cite Kruskal, but it would certainly also be appropriate to cite Chow-Liu.

> I am curious as the core distinctions among the following work and forest density estimation, which serves as the foundation for the submission?

All of these references are relevant to our paper in terms of the use of the Chow-Liu algorithm to build a maximum weight spanning tree. We extend such methods by combining trees with graphs containing cycles, with unrestricted tree-width.

Reviewer 5:

> Why should I do all of this work to learn a blossom-tree model, over a non-paranomral model, a sparse multivariate gaussian or a forest-tree model?

We hope the motivation is clear, that our framework combines the strengths of the nonparanormal (Gaussian graphs) and tree models (nonparametric bivariate edges) in a fairly simple manner. When there is good reason to doubt the joint Gaussian/nonparanormal assumption, this may give a useful approach that is worth the extra work.

> I found the result in Theorem 3.1 a bit confusing. It tells us about the quality of estimating the negentropy. Why is this useful for the joint estimation problem? What does it tell us about the estimates produced using the blossom-tree factorization?

We use negentropy with the motivation of identifying non-Gaussian edges. The theorem indicates when we can correctly identify such edges. It does not directly inform us about the accuracy of the blossom-tree model.

> In addition, please be specific about the complexity of the proposed method, relative to learning using graphical lasso, non paranormal and forest-trees.

If n is the sample size and d is the number of variables/nodes, the complexity of the glasso and nonparanormal estimators is O(n*d^2+d^3). If the bivariate kde is evaluated on the m-by-m grid, the
computational time for forest density estimation is O(n*m^2*d^2). If k is the number of trunks, the blossom tree method scales as O(n*m^2*d^2+k*d^3).

> Calling this a blossom tree is a poor choice. The word blossom was introduced by Edmonds for the matching problem and refers to an odd-sized set of vertices.

Since the two uses of "blossom" appear in such different contexts, this will hopefully not cause any misunderstanding.

> Why are the '0' trunks not equal to the graphical lasso? If you don't add any trunks isn't there just one blossom, modeled as a multivariate gaussian?

'0' trunks does indeed correspond to the glasso. In Figure 2, the first point for the blossom tree refers to the 1-trunk case. We will clarify this in the manuscript.

> Runtimes for the different methods should also be reported.

With n=400 and grid size m=128, when the number of nodes is d=20, 40, and 80, the runtimes on a laptop for the glasso are 1.6s, 5.2s, and 24.9s. The runtimes for the forest are 8.2s, 33.6s, and 136.9s. The runtimes for the blossom tree are 109.9s, 1983.7s, and 25017.7s. The computational cost could be reduced by a C implementation of the algorithms since loops in R are slow.

> The statement "maintain tractable inference without placing limitations on the independence graph" is a bit misleading. They do encourage the graph to be sparse; they are just not structural like, say a graph laplacian.

Agreed. This can be stated more accurately.